# MotionFlow: Learning Implicit Motion Flow for Complex Camera Trajectory Control in Video Generation

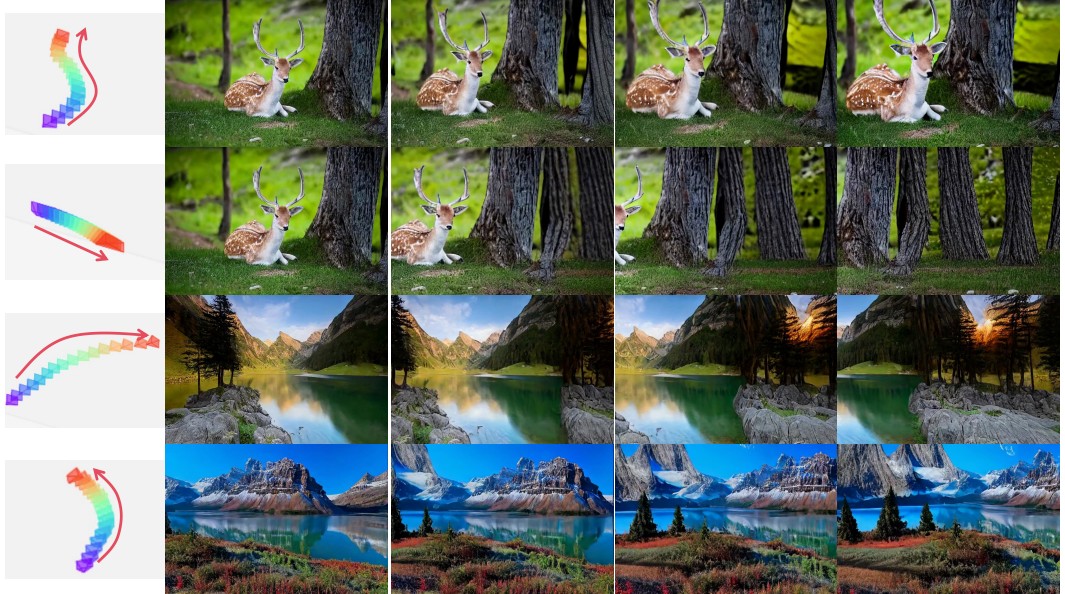

Figure 1: Our method faithfully generates highly realistic and multi-view consistent videos from the given camera trajectory and reference image.

## Abstract

Generating videos guided by camera trajectories poses significant challenges in achieving consistency and generalizability, particularly when both camera and object motions are present. Existing approaches often attempt to learn these motions separately, which may lead to confusion regarding the relative motion between the camera and the objects. To address this challenge, we propose a novel approach that integrates both camera and object motions by converting them into the motion of corresponding pixels. Utilizing a stable diffusion network, we effectively learn reference motion maps in relation to the specified camera trajectory. These maps, along with an extracted semantic object prior, are then fed into an image-to-video network to generate the desired video that can accurately follow the designated camera trajectory while maintaining consistent object motions. Extensive experiments verify that our model outperforms SOTA methods by a large margin. Please visit our anonymous project page to watch the generated videos.

## 1 Introduction

Through extensive and in-depth research on diffusion models(Song et al., 2020; Saharia et al., 2022; Yu et al., 2023), researchers have significantly improved the quality and diversity of video generation. Recent models, such as DynamicCrafter (Xing et al., 2023), exhibit the ability to generate long,

high-quality videos with complex dynamics. However, these methods focus on prompts like texts, images, depth maps, contour maps, human pose or projection maps(Guo et al., 2023a; Hu et al., 2023; Guo et al., 2023b; Li et al., 2023; Mou et al., 2023; Peruzzo et al., 2024; Wu et al., 2023; Zhang & Agrawala, 2023; Hu et al., 2023), often neglecting the precise control over camera motions, which is essential in applications such as film production, virtual reality (VR), and augmented reality (AR). These fields require not only high-quality video contents but also precise adherence to specified camera movements to achieve the desired visual effects.

To enable precise camera control in video generation, several strategies have been developed to integrate trajectory information into generative networks. Although they have made certain progress in improving the flexibility in control and video quality, the generalizability and consistency still need improvement. AnimateDiff (Guo et al., 2023b) aggregate camera information through LoRA (J. et al., 2022) to conditionally guide video generation along specific directions. However, its flexibility is limited when handling user-customized camera trajectories. MotionCtrl (Wang et al., 2024c) utilizes the camera transformation matrix as trajectory information and injects it into the generative model.It employs separate modules to learn camera and object motions, allowing independent motion control in small-scale scenes such as indoor scenes. Nevertheless, Training the two types of motion separately while ignoring their relationship may lead to confusion regarding the relative positions of objects and the scene. Additionally, its relatively simple encoding of camera information restricts its applicability to precise camera control for large-scale outdoor scenes. CameraCtrl (He et al., 2024) adopts plücking embeddings (Sitzmann et al., 2021) as the primary camera trajectory representation, which enhances the implicit mapping between camera motion and pixels. However, the generalizability is still limited, as it struggles to generate videos that differ substantially from the training data. Additionally, these methods primarily support text prompt inputs, which often fail to accurately convey the visual details of the desired video.

To address the above issues, we propose MotionFlow, a novel video generation network guided by camera trajectories, which is capable of producing consistent and coherent 3D videos. Considering the limited visual information provided by text guidance, we instead opt for image guidance to accurately capture the desired video details. Unlike previous methods that modeled camera trajectories and object motions separately to guide the video generation process, we adopt a pre-trained image stable diffusion model that progressively and synchronously interacts with both camera motions and image semantic features. This approach enables us to jointly encode semantic information as object-aware motion priors during video generation. By mapping camera and object motions onto pixel trajectories and training a network to learn these pixel movements, we effectively avoid the confusion associated with separate learning methods.

Our network is built on AnimateDiff (Guo et al., 2023b), a pre-trained image-to-video generation framework, to leverage its high-quality video generation capabilities. To enable precise camera trajectory-guided control and ensure the generalizability of the method across various scenarios, we employ an image stable diffusion network to progressively fuse the features of the reference image with the camera motion trajectory. This information is iteratively injected into the video generation network using cross attention mechanism as a pixel motion prior. Moreover, to directly enhance the geometric consistency of the synthesized videos, we implicitly learn the semantic object features from the reference image and incorporate them into the video generation network through cross-attention mechanism using these features. This approach improves the quality of the synthesized video frames that contain the semantic objects during the iterative denoising steps of the diffusion model. Extensive experimental results demonstrate that our method demonstrates superior 3D consistency, visual quality, and camera trajectory controllability compared to previous approaches.

We summarize our main contributions as follows:

- We introduce a framework that iteratively uses an image diffusion network to learn implicit pixel-level motion flows from camera trajectories and image guidance jointly, achieving high-quality video generation results that adhere to the input camera trajectories.

- We propose to extract semantic object information from the image guidance to enable the video generation network to be aware of these objects through object attention, thereby improving the quality of these objects' pictures in the generated videos.

- We conduct extensive experiments to demonstrate the superiority of our MotionFlow network over SOTA methods both qualitatively and quantitatively.

## 2 RELATED WORKS

### 2.1 CONDITIONAL IMAGE TO VIDEO GENERATION

Conditional image to video (I2V) generation aims to synthesize videos guided by user-provided cues. Recent methods often leverage diffusion models for their stability in training and flexibility in manipulation. Ni et al. (2023) proposes to generate latent flow sequences using a non-pretrained diffusion model to animate images. Their work effectively synthesize motions like facial expression, human actions, and gestures against an almost static background. Shi et al. (2024) introduce a two-stage training approach, each utilizing a diffusion model to generate explicit motion maps and corresponding video. Although these methods could produce vivid motions in open-domain scenarios, they can not incorporate explicit and precise camera control for the video generation task.

### 2.2 VIDEO GENERATION WITH CAMERA CONTROL

To facilitate camera control, most methods typically use text prompts and specific camera movement information to guide the generation of corresponding videos through cross-modal large models.

To effectively control camera motion, AnimateDiff (Guo et al., 2023b) employs the motion LoRA modules to enable specific camera movements, though quantitative control can be challenging. Focusing on the explicit control of both camera and object motion, MotionCtrl (Wang et al., 2024c) achieves flexible motion control. However, its camera motion control module relies on 12 pose matrix parameters as input, which may not sufficiently capture the geometric cues needed for precise camera control. Direct-a-video (Yang et al., 2024) proposes a camera embedder to manipulate camera poses, but it only conditions on some basic camera parameters(such as pan or zoom), limiting its control capabilities. In contrast, CameraCtrl (He et al., 2024) adopt plücker embeddings as the camera trajectory parameters, effectively injecting the camera information. Based on this embedding, VD3D (Bahmani et al., 2024) and CamCo Xu et al. (2024) respectively introduce a ControlNet-like conditioning mechanism and an epipolar attention module to better incorporates camera embeddings. Our method further employs a twin diffusion model to encode these plücker embeddings, allowing for the adaptive and progressive introduction of camera pose information during the iterative denoising process of the diffusion model. Additionally, we enhance plücker embeddings into progressive implicit embeddings, further improving global consistency and stability in the generated videos.

By adopting image prompts, our approach provides a more direct and accurate condition for generation. Moreover, it can be combined with cross-modal large models to facilitate multi-stage, text-driven video generation.

## 3 METHOD

The core idea of our method involves the progressive injection of semantic and pixel-level motion information into a diffusion-based video generation network. As illustrated in Figure 2, we employ a reference motion network to extract pixel-level motion with semantic information from a reference image and a camera trajectory, which serves as a motion flow to guide video generation through reference attention. To further identify potential moving foreground objects and stationary backgrounds within the scene, we introduce a semantic extractor to capture semantic information. This information is then injected into the video generation network through both pixel-wise addition and object attention. For the video generation network, we utilize the pre-trained model of AnimateDiff (Guo et al., 2023b), enhancing it by incorporating reference attention and object attention in each block of the UNet architecture.

This section is structured as follows: Section 3.1 provides a brief overview of the base video generation network; Section 3.2 delves into the camera encoding information; Section 3.3 discusses the integration of semantic information into the video generation backbone network; Section 3.4 explores the incorporation of pixel motion, generated by the reference motion network, into the video generation framework to progressively guide the process; Finally, Section 3.5 presents a detailed analysis of pixel-level motion with semantic and camera motion information.

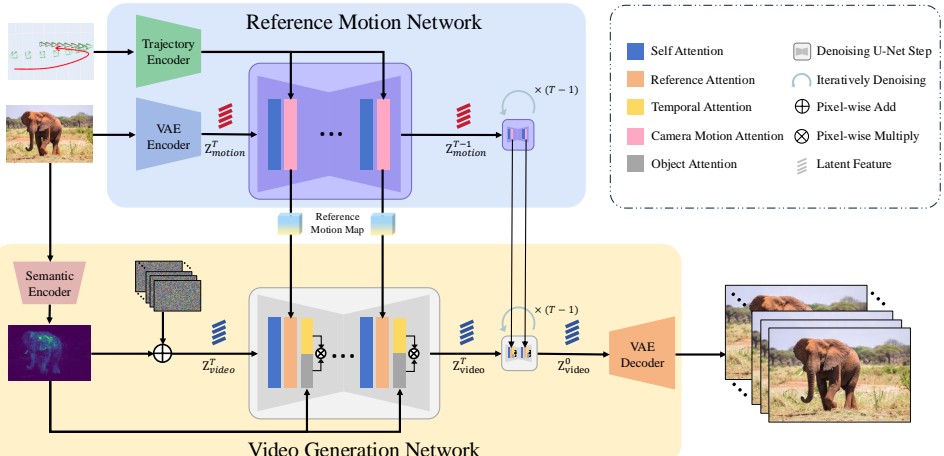

Figure 2: The overview of *MotionFlow*. Our framework is mainly constituted of two parts. In the reference motion network, the camera trajectory is initially encoded using Trajectory Encoder and added to the reference model using camera motion attention with the reference image to get the reference motion priors. For the video diffusion process stage, firstly semantic features are extracted through the semantic encoder to calculate the object attention and also fused with multi-frame noise. Secondly, reference pixel motion is integrated through reference attention. In addition, Temporal Modules are utilized to ensure consistency of generated video.

## 3.1 VIDEO DIFFUSION MODELS

Video diffusion models are a class of generative models that extend the principles of diffusion probabilistic models for image generation to the domain of video synthesis. These models learn to reverse a gradual noising process applied to video sequence data, enabling the generation of high-quality, temporally coherent video sequences. Let $x_0 \in R^{f \times h \times w \times c}$ represent a video latent with $f$ frames, each with dimensions $h \times w$ with $c$ channels. The forward diffusion process is defined as a Markov chain that gradually adds Gaussian noise to the original video:

$$x_t = \sqrt{\bar{a}_t} x_{t-1} + \sqrt{(1 - \bar{a}_t)} \varepsilon, \varepsilon \sim N(0, 1), \tag{1}$$

where $t \in 1, ..., T$ denotes the diffusion step, $\bar{a}_t$ is a noise scheduler and $\varepsilon$ is sampled from a standard Gaussian noise. The model learns to reverse this process, estimating $p(x_{t-1}|x_t)$, typically parameterized by a neural network $\theta$. The objective is to minimize the loss:

$$\mathcal{L}(\theta) = \mathbb{E}_{x_0, \varepsilon, cond, t} \left[ \| \varepsilon - \hat{\varepsilon}_\theta (x_t, cond, t) \|_2^2 \right], \tag{2}$$

where $cond$ represents the conditioning inputs, which may include textual prompts, reference images, camera trajectories, and scene outlines. Video diffusion models incorporate spatiotemporal architectures to capture both spatial details and temporal dynamics. They generally employ mechanisms such as 3D convolutions, attention layers, or recurrent structures to ensure consistency across frames. These models have demonstrated remarkable capabilities in tasks such as video generation, editing, and motion transfer, pushing the boundaries of video synthesis quality and controllability.

## 3.2 TRAJECTORY ENCODER

The trajectory of camera motion can be represented through the transformation of the camera between various frames within a video sequence. This transformation can be succinctly described using the rotation matrix $\boldsymbol{R} \in SO(3)$, and the translation matrix $\boldsymbol{t} \in \mathbb{R}^3$. MotionCtrl (Wang et al., 2024c) represents camera motion by flattening camera parameters into 12D vectors and concatenating them with time embeddings from the base video generation network. A lightweight fully connected network then ensures dimensional consistency between the new combined embeddings and the original time embeddings. However, this representation has limitations in accurately describing camera motion for two primary reasons: 1. Camera parameters inherently reflect characteristics of the camera space,

where rotation matrices and translation vectors carry distinct semantic meanings and should be treated separately rather than flattened into a single vector. 2. Modeling the correlation between raw camera parameters and pixel space is challenging, potentially limiting the model's generalization capacity, particularly for large-scale complex outdoor scenes. In order to better describe the camera pose, we use plücker embeddings (Sitzmann et al., 2021) as the representation of camera trajectory. Given the extrinsic and intrinsic camera parameters $\mathbf{R}, \mathbf{t}, \boldsymbol{K}_f$ for the $f$-th frame, we derive a plücker embedding $\ddot{\boldsymbol{p}}_{f,h,w} v \in \mathbb{R}^6$ for each pixel located at $(h, w)$. This embedding represents the vector from the camera center to the pixel's position as:

$$\ddot{\boldsymbol{p}}_{f,h,w} = \left(\boldsymbol{t}_f \times \hat{\boldsymbol{d}}_{f,h,w}, \hat{\boldsymbol{d}}_{f,h,w}\right), \quad \hat{\boldsymbol{d}}_{f,h,w} = \frac{\boldsymbol{d}}{\|\boldsymbol{d}_{f,h,w}\|}, \quad \boldsymbol{d}_{f,h,w} = \boldsymbol{R}_f \boldsymbol{K}_f [w, h, 1]^\top + \boldsymbol{t}_f \quad (3)$$

Computing plücker embedding for each pixel results in a representation $\ddot{\boldsymbol{P}} \in \mathbb{R}^{6 \times F \times H \times W}$ for a specified trajectory. To inject the trajectory representation into the reference motion network, we designed a trajectory encoder structurally similar to the camera encoder in CameraCtrl (He et al., 2024). However, we improved the architecture: after each 2D ResNet block, we replaced the temporal attention with self-attention and output multi-scale trajectory features.

### 3.3 SEMANTIC ENCODER

We design a semantic encoder to extract the semantic features of salient objects to let the video generation network be aware of these objects. To achieve this, one feasible way is to mark the area of these objects and send it to object attention modules, as shown in the Figure 2. To get these fine-grained information, our semantic encoder employs a lightweight ViT architecture (Caron et al., 2021) to adaptively identify potential areas of salient objects. This module is trained in the first stage, which will be explained in detail in Section 3.6.

### 3.4 REFERENCE MOTION NETWORK

Recent works mainly rely on textual prompts combined with camera features for the camera trajectory control in video generation (He et al., 2024; Wang et al., 2024c). However, due to the loss of scene details in the prompts, it is difficult for them to accurately learn the features to represent the motion of foreground objects and background scenes, potentially leading to low-quality results. Therefore, images are often preferred in I2V tasks since they can convey more fine-grained scene details compared to textual prompts. Some methods (Chen et al., 2023; Xing et al., 2023) adopt CLIP image features in place of CLIP text features and feed them into the diffusion network through cross-attention to achieve image guidance. However, the CLIP features emphasize the text-image alignment and are learned with low-resolution(224×224) image input, which still prioritize high-level abstraction of images and ignore image details.

Another approach is to leverage the ControlNet architecture (Zhang et al., 2023) to achieve camera trajectory control. While this approach can utilize different conditional information to guide video generation, its key limitation is that the supplementary conditional information must be well-aligned with the underlying image features to enable effective interaction. Yet, the trajectory information in camera space and the features in image space exist in different domains, making it challenging for ControlNet to directly convert the trajectory information into motion guidance that the video generation network can understand. Furthermore, the lack of high-quality video data that contains precise camera poses also hinders the generalization capability of such approaches, as discussed in Section 4.2,.

Our idea is to introduce a separate reference motion network to integrate the camera motion trajectory and the semantic information of the reference image, directly producing reference motion maps that can be used in the image space. Specially, as shown in Figure 2, this network adopts a pretrained image stable diffusion(SD1.5 Unet [1]) architecture (Voleti et al., 2024) that incorporates both self and cross-attention layers within the spacial transformer. We inject the multi-scale camera features into

---

[1]https://huggingface.co/stable-diffusion-v1-5/stable-diffusion-v1-5

the cross-attention layers to effectively capture semantic pixel motions as

$$F_{\text{out}} = \text{Softmax}\left(\frac{QK_i^T}{\sqrt{d}}\right)V_i + \lambda_c \cdot \text{Softmax}\left(\frac{QK_c^T}{\sqrt{d}}\right)V_c, \qquad (4)$$

where $Q = \varphi_i(z_t)W_z$, with $\varphi_i(z_t) \in \mathbb{R}^{N \times d_\epsilon^i}$ representing the spatially flattened tokens of the video latent. Here, $K_i = \Psi(img)W_{ki}$ and $V_i = \Psi(img)W_{vi}$, where $\Psi$ denotes the CLIP image encoder. Similarly, $K_c = \Phi(cam)W_{kc}$ and $V_c = \Phi(cam)W_{vc}$, with $\Phi$ representing the Trajectory Encoder. The parameter matrices is denoted by $\mathbf{W}$, and $\lambda_c$ is the coefficient that balances image condition and camera condition.

The weighted combination of camera and image conditions in the cross-attention allows the reference motion network to integrate the encoded camera trajectory with the initial image features effectively. Through end-to-end training, the network is able to predict reference motion maps that contain pixel-level motion information caused by camera pose changes in a frame-by-frame manner. These maps are then injected into the base video generation network through reference attention, as detailed in section 3.5

### 3.5 Video generation Network

We adopt AnimateDiff (Guo et al., 2023b) as the foundation of our video generation network. We utilize the camera-related frame-by-frame reference motion maps generated by the reference motion network to guide pixel-level motion in the video generation process. Specifically, for each block of the video generation network, given the feature map $m_i \in \mathbb{R}^{f \times c \times h \times w}$ from the $i$-block of the base network and the pixel motion $p_i \in \mathbb{R}^{f \times c \times h \times w}$ from reference motion network, we concatenate $p_i$ with $m_i$ along the $w$ dimension to obtain $m_i' \in \mathbb{R}^{f \times c \times h \times 2w}$. Then we perform cross-attention and extract the first half along the $w$ dimension of the $m_i'$ as the input of the objection attention, referred to as reference attention. As described in Section 3.3, we subsequently employ the semantic encoder to capture semantic information, addressing potential moving foreground objects and stationary backgrounds. This information is injected into the video generation network through addition and object attention operations, as illustrated in Figure 2. We not only integrate semantic information during the noise input process but also compute an attention map as a semantic mask between the semantic feature map and the output of reference attention, referred to as object attention. Simultaneously, inspired by the architectural concepts of AnimateDiff (Guo et al., 2023b), we apply temporal attention to the output of reference attention to consolidate consistency between consecutive frames. Specifically, we reshape the input feature map $m_i \in \mathbb{R}^{c \times f \times h \times w}$ to the shape $(h \times w) \times f \times c$. Then we calculate the self-attention across the temporal dimension $f$. The final result is then pointwise multiplied with the semantic mask to produce the output of this block.

### 3.6 Training Strategy

The video generation network is initialized using the pre-trained weights from AnimateDiff (Guo et al., 2023b), while the reference motion network is initialized with the pre-trained Stable Diffusion V1.5 (Rombach et al., 2022) model. The semantic encoder is initialized from ViT-S based DINO (Caron et al., 2021). All other modules are initialized to zero. The MotionFlow is trained using the Adam optimizer (Kingma & Ba, 2015), $8\times$ GPUs, and batch size of 1 per GPU. In the first stage, we train the Trajectory Encoder and motion extractor with a learning rate of $1e^{-4}$ for one day, keeping the reference motion network and video generation network fixed. Subsequently, we train all parts for three days with a reduced learning rate of $1e^{-5}$. We use RealEstate10K dataset (Zhou et al., 2018) for training, which consists of $62\,992$ video clips for training and $7\,391$ video clips for testing accompanied by diverse camera trajectories. To further evaluate the generalizability of our model, we test it on the large-scale outdoor dataset DL3DV-10k (Ling et al., 2024), which contains approximately $10\,000$ videos with known camera poses as well as the generation of videos for animals.

## 4 Experiments

Our experiments are conducted on a server equipped with $8\times$ NVIDIA A800 GPUs. To better evaluate the control of camera trajectories, we designed separate comparisons for small-scale and

large-scale camera movements. Specifically, we conducted experiments to compare the geometric consistency of the generated videos, the semantic consistency, and the alignment between the camera trajectories and the target videos. Experimental results also reveal that our method can generate high-quality videos for dynamic scenes that contain the motion of a part of active foreground objects, such as the head movement of an animal.

CameraCtrl (He et al., 2024) and MotionCtrl (Wang et al., 2024c) are three baseline methods that are most relevant to our method. To ensure fair comparisons, we trained all the methods on the RealEstate10K (Zhou et al., 2018) dataset and tested them using the same camera trajectories and image prompts. To validate the generalizability of our model, we compare our model with other image-to-video generation models on DL3DV-10k(Ling et al., 2024), which is new to all the models.

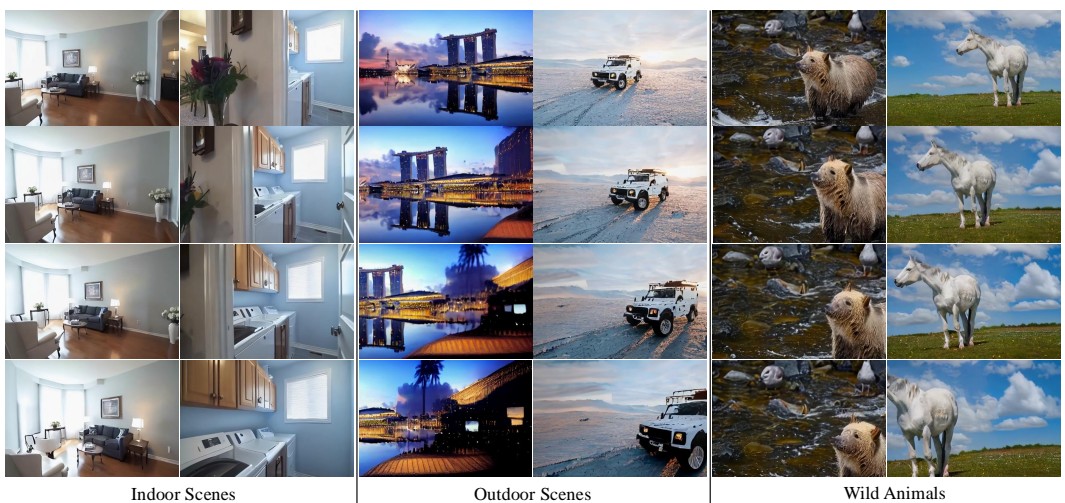

| Indoor Scenes | Outdoor Scenes | Wild Animals |

Figure 3: Qualitative Results. Given a reference image (the topmost image of each group), our approach demonstrates the ability to any camera trajectory. The illustration showcases results with clear, consistent details, and continuous motion.

### 4.1 METRICS

We evaluate our method from two aspects: the alignment of camera trajectories and the visual quality of the generated videos. The distance between predicted and ground-truth camera trajectories is used to measure the alignment. Specifically, we predicted the camera trajectories for both the generated and real videos using the same methods to eliminate the potential scale differences caused by different Structure-from-Motion techniques. Considering the scale and differences between the rotation and translation parameters in the camera matrix, we use *rotation error* and *translation error* to assess them individually, following the approach of Wang et al. (2024c); He et al. (2024).

Four classical image-level quality metrics, including Fréchet Inception Distance (FID) (Heusel et al., 2017), SSIM (Wang et al., 2004), PSNR (Hore & Ziou, 2010) and LPIPS (Zhang et al., 2018), are used to evaluate the quality of the generated video frames, and video-level metric Fréchet Video Distance (FVD) (Unterthiner et al., 2018) is further applied to assess video-level quality, similar to previous video generation methods(Wu et al., 2023; Khachatryan et al., 2023; Cai et al., 2023b; Ceylan et al., 2023; Cai et al., 2023a).

### 4.2 COMPARISON WITH STATE-OF-THE-ART METHODS

**Geometric consistency.** The effectiveness of camera-guided control is evaluated with the camera trajectories estimated with the Structure-from-Motion technique. Due to the low overlap between images of RealEstate10K scenes, the commonly used COLMAP (Schönberger & Frahm, 2016) frequently fails to estimate their corresponding camera trajectories. Instead, we used ParticleSfM (Zhao et al., 2022) similar to MotionCtrl(Wang et al., 2024c), a pre-trained network framework capable of estimating camera trajectories in complex and dynamic scenes, to get the camera trajectories of generated videos. We evaluate the rotation part $R \in \mathbb{R}^{3 \times 3}$, and the translation part $T \in \mathbb{R}^{3 \times 1}$ separately.

To fully compare our model and other camera trajectory guided video generation models, we conduct basic trajectory (sample every 8 frames) and difficult trajectory (sample every max frame we can sample). To guarantee the reliability of the experiment, we evaluated the error between the generated video's camera trajectory and the ground truth (GT) video trajectory using ParticleSfM (Zhao et al., 2022), Dust3R (Wang et al., 2024b) and VggSfM (Wang et al., 2024a). As shown in Table 1, our method can generate videos that align better with the GT camera trajectories compared to other methods, both in basic and difficult trajectories.

Table 1: Quantitative comparisons (Pose got by Dust3r, VggSfM, and ParticleSfM). We compare against prior work on basic trajectory and random trajectory respectively. T-Err and R-Err, representing TransErr and RotErr respectively, are reported as the metrics from Appendix A.

| | Basic Trajectory | | | | | | Difficult Trajectory | | | | | |
| | Dust3R | | VggSfM | | ParticleSfM | | Dust3R | | VggSfM | | ParticleSfM | |
| | T-Err↓ | R-Err↓ | T-Err↓ | R-Err↓ | T-Err↓ | R-Err↓ | T-Err↓ | R-Err↓ | T-Err↓ | R-Err↓ | T-Err↓ | R-Err↓ |
|---|---|---|---|---|---|---|---|---|---|---|---|---|
| CameraCtrl | 0.092 | 0.300 | 1.499 | 0.204 | 2.80 | 0.924 | 0.080 | 0.291 | 1.711 | 0.195 | 3.17 | 0.648 |
| MotionCtrl | 0.059 | 0.228 | 0.882 | 0.215 | 2.02 | 0.914 | 0.054 | 0.227 | 1.012 | 0.226 | 2.10 | 0.739 |
| Ours | 0.043 | 0.223 | 0.767 | 0.156 | 1.66 | 0.886 | 0.053 | 0.210 | 0.802 | 0.146 | 1.77 | 0.574 |

**Visual Quality.** The image-level and video-level quality metrics are adopted for the video consistency and image quality evaluation. To ensure fair evaluation, all models for comparison are trained on the same dataset RealEstate10K with the same baseline SD1.5. As the the experimental results demonstrated in Table 2 and Figure 4, our model not only generates high-quality videos guided by camera trajectories but also outperforms other methods of the same type in terms of video generation quality, highlighting the effectiveness of our approach. In Figure 1 and Figure 3, we further demonstrate the results of videos generated by our method in indoor, outdoor, and dynamic scenes, showcasing its adaptability and robustness across various video generation tasks.

Table 2: Quantitative comparison on visual quality.

| | Basic Trajectory | | | | | Difficult Trajectory | | | | |
| | LPIPS↓ | PSNR↑ | SSIM↑ | FID↓ | FVD↓ | LPIPS↓ | PSNR↑ | SSIM↑ | FID↓ | FVD↓ |
|---|---|---|---|---|---|---|---|---|---|---|
| CameraCtrl | 0.791 | 8.62 | 0.212 | 99.83 | 1079 | 0.796 | 7.57 | 0.17 | 112.77 | 1023 |
| MotionCtrl | 0.732 | 9.48 | 0.266 | 81.28 | 972 | 0.728 | 8.70 | 0.24 | 84.68 | 789 |
| Ours | 0.206 | 17.5 | 0.567 | 38.48 | 348 | 0.255 | 17.7 | 0.54 | 41.93 | 390 |

**Generalizability Comparison.** The outdoor dataset DL3Dv datasets (Ling et al., 2024) is employed to evaluate the generalizability of models which are trained solely on indoor datasets. As shown in Table 3, our method demonstrates superior image-level and video-level quality with SOTA methods (Xing et al., 2023; Chen et al., 2023; Guo et al., 2023b). This highlights the enhanced robustness and generalizability of our method, which is attributed to the additional diffusion model that progressively facilitates the synchronization of camera pose with the reference image. As a result, it not only improves the original video's generation capability but also provides better guidance and control. The specific results are shown in Figure 5 and Table 3.

Table 3: Generalization ability of outdoor scenes. IC: image condition.

| | Real10k | | | | | DL3DV | | | | |
| | LPIPS↓ | PSNR↑ | SSIM↑ | FID↓ | FVD↓ | LPIPS↓ | PSNR↑ | SSIM↑ | FID↓ | FVD↓ |
|---|---|---|---|---|---|---|---|---|---|---|
| AnimateDiff | 0.326 | 15.22 | 0.478 | 55.43 | 508 | 0.632 | 11.04 | 0.239 | 64.87 | 793 |
| VideoCrafter | 0.573 | 11.81 | 0.338 | 60.49 | 708 | 0.529 | 12.45 | 0.280 | 83.35 | 901 |
| DynamicCrafter | 0.272 | 16.38 | 0.523 | 47.91 | 589 | 0.336 | 14.92 | 0.339 | 70.12 | 719 |
| MotionCtrl | 0.763 | 9.14 | 0.218 | 74.84 | 977 | 0.776 | 9.77 | 0.174 | 97.54 | 1116 |
| CameraCtrl+IC | 0.479 | 13.62 | 0.423 | 60.89 | 704 | 0.554 | 12.87 | 0.282 | 85.21 | 1006 |
| Ours | 0.188 | 18.70 | 0.599 | 22.55 | 311 | 0.282 | 15.64 | 0.368 | 55.62 | 505 |

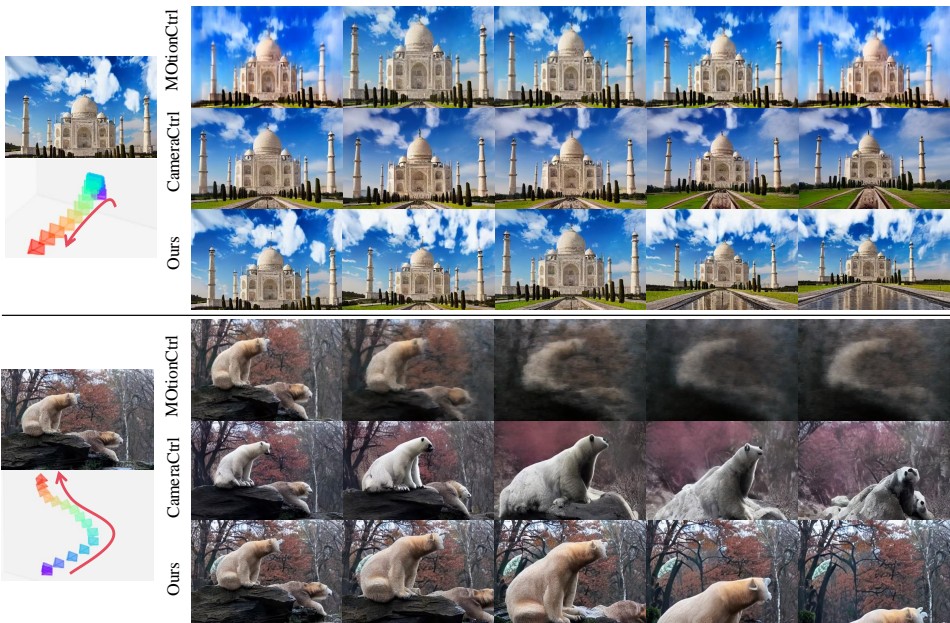

Figure 4: Qualitative comparison on the control ability of and video quality.

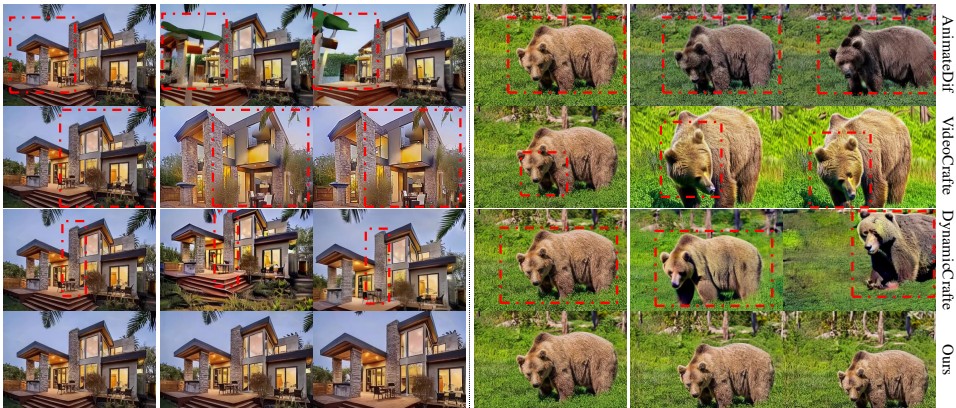

Figure 5: Qualitative comparison on the generalization ability of outdoor scenes. Cross-frame misalignment and artifacts are highlighted with red dashed boxes.

### 4.3 ABLATION STUDY AND ANALYSIS

**Reference Motion Network.** We first verified the effectiveness of the reference motion network, the most important module in our model. The first line of the Table 4 shows results generated with the network trained without this network, where the output of trajectory encoder is directly fed into the reference attention of video generation network. Significant drops of all metrics can be observed when training without reference motion maps to provide reference motion flow. We further explored the benefit of using stable diffusion pre-trained parameters for the reference motion network. The results shows that adopting pre-trained parameters leads to an overall improvement, especially the generative quality. This improvement can be attributed to the pre-trained model's strong 3D priors and image generation capabilities.

**Semantic Extractor.** As the second line of Table 4, the semantic extractor results in an overall improvement. This is because the semantic extractor provides a semantic prior of the reference image to the video generation network, which contributes to the decoupling of various objects with complex motion.

Table 4: Ablation study.

| | Visual Quality | | | | | Camera Trajector Alignment | |
|---|---|---|---|---|---|---|---|
| | LPIPS↓ | PSNR↑ | SSIM↑ | FID↓ | FVD↓ | TransErr↓ | RotErr↓ |
| w/o referNet | 0.281 | 16.58 | 0.521 | 41.93 | 463 | 1.12 | 0.297 |
| w/o Semantic Encoder | 0.194 | 18.23 | 0.561 | 25.69 | 319 | 0.697 | 0.296 |
| w/o stable diffusion pretrain | 0.218 | 16.87 | 0.532 | 39.53 | 393 | 0.722 | 0.299 |
| Full Model | 0.188 | 18.70 | 0.599 | 22.55 | 311 | 0.608 | 0.295 |

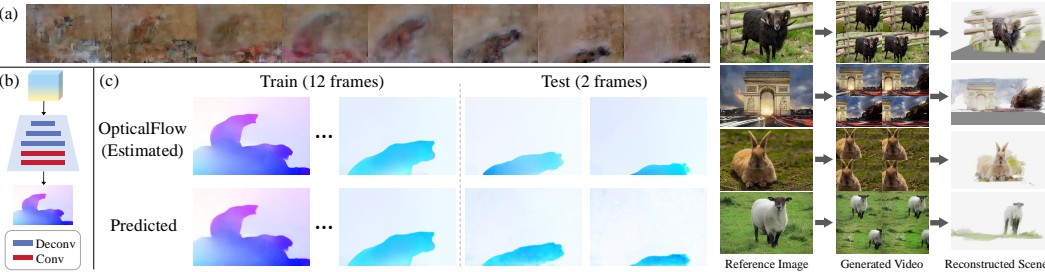

Figure 6: Toy experiment for the Reference Motion Maps interpretation.

Figure 7: Scene Reconstruction with generated video.

**Analysis of Reference Motion Maps.** After translating reference motion maps (RMMs) into images through the decoder of the pre-trained VAE, we observe that these images seem to record the salient object movements in generated videos, as shown in the Figure 6 *(a)*. We thus hypothesize that RMMs contain optical flow (OF) information, and conduct a toy experiment to verify it. First, we estimated OFs from a generated video (de Armas, 2019), and then treated them and the corresponding RMMs as data pairs. Here we choose the video shown in the second example of Figure 5 as an example, where the first 12 pairs of RMMs and OFs are taken as the training set and the 2 pairs at the last two frames as the test set. Second, a shallow network is adopted to translate the RMMs to OFs. The network consists of 3 deconvolution (Noh et al., 2015) layers for decoding features to image resolution and 2 convolution layers for feature translation, as shown in Figure 6 *(b)*. We trained the network for 1k epochs with a batch size of 12, L1 loss, and the Adam optimizer, taking approximately 40 seconds. As the results illustrated in Figure 6 *(c)*, the OF information can be directly extracted from RMMs of test frames. While this experiment verifies the correlation between RMMs and OFs, we argue that the learned RMMs have more abundant information than OFs to facilitate the camera trajectory control.

### 4.4 APPLICATION: 3D SCENE GENERATION

Our method not only retains the original video generation capability of the model but also incorporates camera control. With only a single reference image and a panoramic camera trajectory, we can generate a corresponding scene video and can further perform 3D reconstruction on this video to obtain explicit 3D scenes, as shown in Figure 7.

### 5 CONCLUSION, LIMITATION, AND FUTURE WORK

Through a comparative study with existing camera-guided video generation frameworks, such as CameraCtrl and MotionCtrl, our method demonstrates superior performance in controlling camera movements across different scales. Furthermore, our model also demonstrates a significant advantage in image and video quality compared to other video generation approaches.

Although our method can generate high-quality dynamic scenes while considering potential object motion during camera movement, it lacks explicit guidance for object motion control. In the future, we plan to employ image or point-based motion guidance to precisely control the spatial object movement and the changes over time, while maintaining the generation quality and camera trajectory alignment.

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

## A    METRICS OF THE CAMERA TRAJECTORY CONTROLLABILITY

Rotation Error: The relative rotation distances are then converted to radians, and we sum the total error of all frames,

$$R_{\mathbf{err}} = \sum_{\mathbf{i=1}}^{\mathbf{n}} \mathbf{arcos}(\frac{\mathbf{tr}(R_{\mathbf{out_i}}^{\mathbf{T}} R_{\mathbf{gt_i}}^{\mathbf{T}}) - \mathbf{1}}{\mathbf{2}}) \tag{5}$$

Translation Error: The norm of the relative translation vector for each frame is also summed together to form the translation error of the whole video,

$$T_{\mathbf{err}} = \sum_{\mathbf{i=1}}^{\mathbf{n}} \|T_{\mathbf{out_i}} - T_{\mathbf{gt_i}}\|_{\mathbf{2}} \tag{6}$$

## B    ETHICS STATEMENT

We strongly discourage the misuse of generative AI to create content that causes harm or spreads misinformation. Our camera trajectory-driven video generation approach could be abused to create misleading or invasive content, especially when fed malicious reference images. To mitigate these risks, we adhere to a strict code of ethics, respect privacy rights, comply with legal standards, and encourage positive adoption of our technology. We also recommend incorporating content safety mechanisms for the input image and generated video to prevent potential misuse, and we promote responsible use of the generated content.

