# OpenReview forum: "MOTIONFLOW:Learning Implicit Motion Flow for Complex Camera Trajectory Control in Video Generation"
_ICLR.cc/2025/Conference — ICLR 2025 Conference Withdrawn Submission_

### Official Review · Reviewer_2Nwz · 2024-10-31

**Soundness:** 2
**Presentation:** 3
**Contribution:** 2
**Rating:** 5
**Confidence:** 3

**Summary:**

The paper introduces MotionFlow, a video generation model designed for multi-view video synthesis with control over camera trajectories and object motion. Unlike prior approaches that treat camera and object motions separately, MotionFlow integrates them by converting both into pixel motion. Key components of the model include the Reference Motion Network, which guides the video generation process by aligning camera trajectory with pixel movements, and the Semantic Encoder, which ensures moving objects maintain coherence across frames. Controlled video generation could be useful for applications in 3D reconstruction, film production, VR, and AR.

**Strengths:**

The paper addresses a gap in video generation by focusing on complex camera trajectory control alongside object motion, which is a requirement for applications in film production, VR, and AR. The approach of combining camera and object motion into pixel motion is more useful than previous separate-learning approaches.

The description of the proposed MotionFlow model, its Reference Motion Network, and Trajectory and Semantic Encoder components is clear. The paper explains the modifications over existing frameworks- AnimateDiff and Stable Diffusion.

The experiments section is good, with a comparison against state-of-the-art methods (CameraCtrl, MotionCtrl).

**Weaknesses:**

It seems that the main novelty of the paper is in the reference motion network , but it is not clear whether this has been completely proposed from the authors or are their other papers that have been used as a baseline.

Although well-explained, the Trajectory Encoder and Semantic Encoder sections could use an expanded diagram of these components in Figure 2, or even a separate figure, for clarity of how these integarte and contribute to the overall network.

The introduction could frame the significance of the problem more compellingly, particularly for applications where camera control is crucial, such as interactive media and virtual environments. This might better capture the impact of the contribution.

The metrics are comprehensive, but adding a brief explanation of why each metric is important to the discussion could improve clarity. For instance, briefly discussing how Rotation Error and Translation Error relate to real-world camera control applications would make the results more meaningful.

Additional qualitative comparisons with other methods would be beneficial, particularly if the authors could include more challenging scenarios, such as multiple moving objects or varying light conditions. Highlighting specific failure cases or limitations of the model, perhaps as a "Limitations" subsection, would enhance transparency.

**Questions:**

Please see above.

---

> ### Author Response · Authors · 2024-11-24
>
> Thank you for your detailed and thoughtful suggestions. We will incorporate these in revision. Specifically:
> 1. We will include additional diagrams to illustrate the detailed processes of the Trajectory Encoder and Semantic Encoder in Figure 2, making them easier to understand.
>
> 2. In the introduction, we will use concrete applications to provide a more intuitive understanding of the paper’s contributions, such as highlighting its potential in VR.
>
> 3. We will briefly explain the specific metrics and include comparative visualizations alongside corresponding metrics to clearly demonstrate the focus of these evaluations.
>
> 4. We will present examples of more challenging scenarios in the limitations section to provide readers with tangible cases and encourage further improvements based on these.
>
> Additionally, we deeply appreciate the time and effort you invested in reviewing our work.

---

### Official Review · Reviewer_JUha · 2024-11-01

**Soundness:** 2
**Presentation:** 2
**Contribution:** 2
**Rating:** 3
**Confidence:** 4

**Summary:**

The paper introduces MotionFlow, a camera-controllable image-to-video (I2V) model. Technically, MotionFlow first utilizes a reference motion network, which takes a reference image and camera trajectory as input to obtain a feature map of the video motion. Next, to incorporate the image condition into the T2V model, the authors employ a semantic encoder to extract semantic features. These semantic features, along with the motion feature map, are injected into a video diffusion model to generate the desired video that accurately follows the designated camera trajectory. Extensive experiments conducted on the RealEstate10K and DL3DV-10ks datasets demonstrate the effectiveness of MotionFlow for precise camera control.

**Strengths:**

S1: The paper proposes a novel feature extraction and injection method for video generation models to enable precise control of camera trajectories (e.g., using a reference motion network to extract motion maps).

S2: The paper presents good qualitative and quantitative results.

**Weaknesses:**

W1: The expression in this paper appears somewhat informal, which may lead to inconvenience and confusion for readers. Here are a few examples, though not exhaustive: for instance, the phrase "Conference submissions" is inadvertently included in the fifth line of the title; there are missing references in lines 135 and 315; a new term, "motion extractor," is introduced in line 316, yet this module is not discussed elsewhere in the paper. Furthermore, the experimental settings are explained in both Section 3.6 and Section 4, which may seem somewhat redundant.

W2：Based on the abstract (lines 47-51), my initial understanding was that the extracted object motion from existing videos would be used, and the generated video would "follow the designated camera trajectory while maintaining consistent object motions." However, upon reviewing the technical details, it seems that the approach essentially uses a reference image as a condition, effectively turning the T2V model into an I2V model. There is no actual extraction of motion priors as implied, which makes this claim somewhat overstated and potentially confusing for readers. Similar statements appear in line 80 ("with the reference image to get the reference motion priors") and line 243 ("The goal is to generate the scene according to the specified camera trajectory while preserving the original motion trajectories of the moving objects").

W3: In line 300, it is mentioned that the semantic information obtained from the semantic encoder can address potential moving foreground objects and stationary backgrounds. I do not fully understand how this works and would appreciate a more detailed and clearer explanation from the authors.

W4: The experimental details in Tables 1, 2, and 3 should be more comprehensive. For example, it would be helpful to specify the number of videos used for evaluation and the principles applied to classify trajectories as basic or difficult.

W5: Based on the results in Table 1, the reported values seem to differ from those in the original CameraCtrl paper. While this paper evaluates basic and difficult trajectories separately, it might be useful to provide results that align more closely with CameraCtrl for a clearer comparison.

**Questions:**

Q1: I hope the authors can address the concerns raised in the "Weaknesses" section.

---

> ### Author Response · Authors · 2024-11-23
>
> Ques: About some detailed issue.
> 1、Our primary contribution focuses on addressing the challenges currently faced by the research community. We provide a detailed explanation of how we resolved these issues and how our method was implemented. However, we acknowledge that some minor details in the paper were not handled as thoroughly as they could have been, and we sincerely apologize for this. We have now uploaded the updated version.
>
> Ques: About object motion.
> 2、The main goal of our paper is to generate a video corresponding to a specified camera trajectory based on reference images while preserving the original motion of objects. Specifically, we achieve this by extracting semantic information from the reference images using a semantic encoder. This semantic information is injected into the Diffusion module's U-Net via a mask. The mask is designed to compute attention only on the background during the reference attention step. However, the overall coherence of the video is ensured through temporal attention across frames. In essence, camera motion only alters the pixel information of the background, leaving the object pixels unaffected. This approach enables separate control over the motion of foreground objects and stationary backgrounds.
>
> Ques: About experiment parts of camera trajectory.
> 3、for the trajectories as basic or difficult, we have explained these parts in our Geometric consistency parts as "basic trajectory (sample every 8 frames) and difficult trajectory (sample every max frame we can sample)"
>
> Ques: About experiment parts of video quality.
> 4、We follow the same way of MotionCtrl, sample 500 videos for basic, sample 500 videos for difficult with total 1000videos.
>
> Ques: About the results  different from original CameraCtrl paper.
> 5、We have tried to align our result with the results in CameraCtrl. But it is a pity that CameraCtrl uses the Colmap the estimate the camera pose of the generated videos while we use other mehtods because Colmap achieved only about a 10\% success rate when tested in real-world scenarios with 16 frames and most of the case, it will failed.(We have confirmed this point with the original authors of CameraCtrl).

---

> > ### Comment · Reviewer_JUha · 2024-11-26
> >
> > The authors state: "The main goal of our paper is to generate a video corresponding to a specified camera trajectory based on reference images while preserving the original motion of objects." However, I did not observe the movement of objects or the preservation of the original motion of objects in the paper. Similarly, the authors frequently emphasize "moving objects" in the paper, but it is clear that most cases involve stationary objects with the camera moving. Reviewer AefY also pointed out this issue.

---

### Official Review · Reviewer_UeMD · 2024-11-01

**Soundness:** 2
**Presentation:** 2
**Contribution:** 3
**Rating:** 5
**Confidence:** 4

**Summary:**

This paper presents a framework for generating videos guided by camera trajectories, addressing the significant challenges of achieving consistency and generalizability, especially when both camera and object motions are involved. To tackle this, they propose an integrated approach that combines camera and object motions by translating them into corresponding pixel movements. Key techniques include a reference motion network that learns the reference motion map aligned with specified camera trajectories, and an object motion prior that aids in maintaining consistent object motions. This information is then utilized by the video generation network to produce videos that accurately follow designated camera trajectories.

**Strengths:**

1.	To introduce camera trajectory control information while avoiding the confusion between camera motion and object motion, this paper proposes a method that adopts a Reference Motion Network that progressively and synchronously interacts with both camera motions and image semantic features. These interactions are then injected into the main denoising network. In contrast to other methods that simply fuse camera trajectory features with the latent space of the denoising network, this approach achieves more effective camera control.

2.	The experimental results demonstrate that this method significantly improves camera trajectory control capabilities compared to other methods.

**Weaknesses:**

1.	Since it is mentioned in line 73 of the introduction that MotionCtrl has a problem: “Training the two types of motion separately while ignoring their relationship may lead to confusion regarding the relative positions of objects and the scene,” and in line 78, it is stated that CameraCtrl's “generalizability is still limited, as it struggles to generate videos that differ substantially from the training data,” visualizing these issues would enhance the persuasiveness of the argument.

2.	For the TRAJECTORY ENCODER in the proposed method, it would be beneficial to explicitly state that the same camera trajectory representation approach as CameraCtrl is used, rather than simply mentioning, "In order to better describe camera pose, we use xxx." Similarly, for the REFERENCE MOTION NETWORK, it should be clearly indicated that a UNet is employed as the reference model, as in ReferenceNet in AnimateAnyone, even though the trajectory encoder has also been integrated.

3.	How the reference network is implemented during the T-step denoising process is not clearly explained. Is the reference image added noise at each time step t and then denoised? This part seems to be unclear in the paper, and it would be best to include some formulas for better description, as well as for the Object Attention section.

**Questions:**

None

---

> ### Author Response · Authors · 2024-11-23
>
> 1、Thank you for your detailed and thoughtful suggestions. We will incorporate these in the updated version.
>
> Ques: about cameractrl struggles to generate videos that differ substantially from the training data
> 2、The can be visualized from the Figure.4. When generating video in outdoors or with animals, the video will become unstable. Specifically, as the camera trajectory moves, the shape or color of the animals may change, and the structure of the scene may get changed. As shown in Figure 4, the color of the bear in the video generated by Cameractrl gradually changes, and the swimming pool in front of the white building gradually turns into grassland. The same problem happened to MotionCtrl.
>
> Ques: About T-step denoising problem
> 3、As no noise was added to the reference Motion NetWork, so at each t-step denoising process, all the timestep will be assume as zero time step during the reference Motion Netwok.
>
> Ques: About Object attention.
> 4、The Object Attention was just denoted as the semantic mask mainly used to distinguish between foreground and background. This part is relatively easy but effective. We just calculate the attention map with the semantic information of the image and the output of reference attention, which then used as mask (set foreground object 0) to calculate pixel-wise multiply with the result of temporal attention detailed in Fig.2. It should be noted that there will be a residual connection in the temporal block designed in original UNet structure. This operation can ensure reference attention only affect the background and the foreground object will keep its original motion( still or moving).

---

### Official Review · Reviewer_AefY · 2024-11-03

**Soundness:** 2
**Presentation:** 2
**Contribution:** 2
**Rating:** 3
**Confidence:** 5

**Summary:**

The paper presents MotionFlow, a camera-controllable image-to-video (I2V) model.
Specifically, MotionFlow uses a reference motion network to process a reference image and camera trajectory, generating motion feature map. A semantic encoder extracts semantic features, which are combined with the motion features and fed into a video diffusion model to generate videos that follow the specified camera trajectory. Experiments on the RealEstate10K and DL3DV-10ks datasets show MotionFlow’s effectiveness in precise motion control.

**Strengths:**

S1: The paper introduces a framework that iteratively uses a Reference Motion Network to guide the Video Generation Network for accurate camera control.
S2: Experiments demonstrate the model's effectiveness.

**Weaknesses:**

W1: The expression in the paper does not appear sufficiently formal, as evidenced by the use of "Conference submissions" in the title and the missing references in line 135.

W2：Both MotionCtrl and CameraCtrl utilize SVD (which generally ensures strong frame consistency) as the foundational architecture for camera control in video generation. However, based on the experimental results in Figure 4, I observed notable distortion in MotionCtrl and temporal inconsistencies in CameraCtrl, which diverge from my expectations. Could the authors please clarify these discrepancies?

W3: In Figures 1 and 3, it appears that most cases illustrate camera movement around static objects rather than emphasizing object motion, as stated in the abstract.

**Questions:**

Q1: This paper uses AnimateDiff as the foundational architecture. I’m curious why the authors chose not to directly use an existing I2V model, such as SVD, for this task. Additionally, when incorporating the reference image as a condition for AnimateDiff, I would like to know why the semantic features are added to the input noise. What is the motivation behind this design choice? Typically, in models like SVD, the reference image latent code is concatenated with the noise latent.
Q2: I hope the authors can address the concerns raised in the "Weaknesses" section.

---

> ### Author Response · Authors · 2024-11-22
>
> 1、Our primary contribution focuses on addressing the challenges currently faced by the research community. We provide a detailed explanation of how we resolved these issues and how our method was implemented. However, we acknowledge that some minor details in the paper were not handled as thoroughly as they could have been, and we sincerely apologize for this. We have now uploaded the updated version.
>
> Ques:Why we choose AnimateDiff as the baseline
> 2、Our method is general and can be applied to various diffusion-based approaches, such as AnimateDiff, SVD etc. Since CameraCtrl and MotionCtrl were t2v-SVD-based methods driven by camera trajectories, for fair comparison, we use the same baseline as the comparative experiments to align the video generation ablity.
>
> Ques: why the semantic feature was added to the noise
> 3、semactic features are added to the noise because it can avoid the change of the latent shape and keep the network align with the original backbone network. Actually, this way is more common.
>
> Ques:You observed notable distortion in MotionCtrl and temporal inconsistencies in CameraCtrl.
> 4、Most of the time, when introducing additional control signals, the original video generation ablity will be influenced a lot.
>
> Ques: It appears that most cases illustrate camera movement around static objects rather than emphasizing object motion.
> 5、Most of the time, it is hard to get the camera pose in dynamic scene. The object motion in our paper means that most of the time, the object behaves normal. We emphasize the camera trajectory of the static background and the foreground object keep its original motion(still or moving). We can ensure that the subject's consistency during the video generation process, and avoid occurrence where animals suddenly move forward, shift to the right, or abruptly disappear.

---

> > ### Comment · Reviewer_AefY · 2024-12-02
> >
> > Reply – Why we chose AnimateDiff as the baseline, and notable distortion in MotionCtrl and temporal inconsistencies in CameraCtrl:
> > I think intentionally degrading the performance of the baseline model (for example, replacing the SVD of MotionCtrl and CameraCtrl entirely with AnimateDiff) does not provide a fair comparison. This concern was also raised in Q1.
> > In my view, a fair comparison would involve using the same base model for both the proposed method and the baseline.
> >
> > Reply – Why the semantic feature was added to the noise:
> > In my experience, it is more common to add the image or video latent to the noise, as this better matchs the intermediate state during the original model's training process. I have not seen many cases where semantic features are directly added to the noise. You mentioned that "this way is more common"—could you provide some references or papers where this approach is used?
> >
> > Additionally, you referred to the case of “objects keeping their original motion (still or moving),” but I could not find this in the paper. Could you clarify or direct me to where this is discussed?

---

### Note · Authors · 2024-12-24

**Comment:**

I acknowledge that I and all co-authors of this work have read and commit to adhering to the ICLR Code of Ethics.

**Withdrawal Confirmation:**

I have read and agree with the venue's withdrawal policy on behalf of myself and my co-authors.